# PROBING THE CONTENTS OF TEXT, BEHAVIOR, AND BRAIN DATA TOWARD IMPROVING HUMAN-LLM ALIGNMENT

## ABSTRACT

Large language models (LLMs) are traditionally trained on massive digitized text corpora; however, alternative data sources exist that may help evaluate and improve the alignment between language models and humans. We contribute to the assessment of the role of data sources in human-LLM alignment. Specifically, we present work aimed at understanding differences in the informational content of text, behavior (e.g., free associations), and brain (e.g., fMRI) data. Using representational similarity analysis, we show that word vectors derived from behavior and brain data encode information that differs from their text-derived cousins. Furthermore, using an interpretability method that we term representational content analysis, we find that, in particular, behavior representations better encode certain affective, agentic, and socio-moral dimensions. The findings highlight the potential of behavior data to evaluate and improve language models along dimensions critical for human-LLM alignment.

## 1 INTRODUCTION

Large language models (LLMs) are trained to predict the occurrence of tokens given their context. Research demonstrates that training larger models on more text leads to predictable improvements on this objective and other benchmarks (Kaplan et al., 2020; Hoffmann et al., 2022).

However, optimizing for next-token prediction does not automatically produce models that align well with people's preferences, representations, or judgments. To remedy this (insofar as such alignment is desired), researchers are incorporating more explicit sources of human data into training and evaluation pipelines.

For instance, in addition to the now-popular use of explicit feedback on language model outputs (e.g., via *reinforcement learning from human feedback* or *direct preference optimization* Christiano et al., 2017; Bai et al., 2022), researchers have also been leveraging semantic textual similarity judgments (e.g., Cer et al., 2017, dataset), sentiment judgments (e.g., Socher et al., 2013, dataset), sensorimotor judgments (Kennington, 2021), as well as brain imaging recordings (Toneva & Wehbe, 2019; Hollenstein et al., 2019, see also, github.com/brain-score/language). Not only do these efforts demonstrate improvements in model helpfulness and accuracy, but they may also improve human-model trust and communication (Sucholutsky & Griffiths, 2023; Bansal et al., 2019), as well as make for more predictive and plausible models of human psychology (Binz & Schulz, 2023; Hussain et al., 2024).

Ultimately, it is clear that human-generated data must play a crucial role, both in *measuring and increasing human-model alignment* (henceforth, just *human-model alignment*). However, it remains an open question which *types* of human data should be used, and what the promise of these prospective types may be.

Prospective data for human-model alignment can be grouped into three types (see also Roads & Love, 2023): text, behavior, and brain. Although text has received considerable attention in language modeling (i.e., for pretraining), behavior and brain data have attracted comparatively little. In light of recent large-scale, high-resolution collection efforts (e.g., De Deyne et al., 2019; Jamali et al., 2024), these two data types might hold untapped potential for human-model alignment. Our study

thus seeks to address two research questions: (a) do behavior and brain data encode systematically different information than text, and (b) are these differences useful from the perspective of human-model alignment?

In what follows, we run a representational similarity analysis (RSA) to uncover systematic differences between text, behavior, and brain data (Section 4.1). We then analyze the content of these differences via our *representational content analysis* (RCA, Sections 4.2, 4.3), and end with a discussion of the merits and limitations of our work.

## 2    OUR CONTRIBUTIONS

Our contributions are four-fold. First, we perform a comprehensive comparison of 10 text representations, 10 behavior representations, and 6 brain representations, revealing robust differences between data types (Section 4.1).

Second, we collate the largest (to our knowledge) metabase of predominantly human-rated (behavioral) word properties (*word norms*), Section 3.1), which we call *psychNorms*. The metabase is publicly available at github.com/[ANONYM]/psychNorms (and in the supplementary materials), and reflects over half a century of psycholinguistic research. We hope it will serve as a valuable resource for researchers seeking to measure and interpret language representations along psychologically meaningful dimensions.

Third, leveraging *psychNorms* and linear probes (see, e.g., Belinkov, 2022), we demonstrate how to build interpretable informational content profiles for abstract representations via a novel analysis framework that we call *representational content analysis* (RCA, Section 3.3). By comparing the profiles of different representations, we can provide crucial insight into the *content* of their differences. This could be especially useful for interpreting and navigating discrepancies between the plethora of otherwise opaque representational alignment metrics (Sucholutsky et al., 2023).

Fourth, and most importantly, we show that, despite being trained on orders of magnitudes less data, the behavior representations encode psychological information of equivalent or even superior reach and quality in comparison to their text-based cousins (Sections 4.2, 4.3). This indicates that behavior contains a wealth of highly concentrated psychological information, and is a powerful complement to text for measuring and improving human-LLM alignment.

We view our work as foundational with respect to the entitled goal of improving human-LLM alignment. By carrying out the necessary groundwork looking into the space of possible data sources and the kinds of information they encode, we hope to pave the way for future researchers seeking to measure and improve the human-likeness of the current state-of-the-art (SOTA).

## 3    METHODOLOGY

### 3.1    REPRESENTATIONS AND NORMS

Our analyses seek to answer (a) whether brain and behavior data offer systematically different information than text, and (b) whether these differences are useful from the perspective of human-model alignment. We attempt to answer these questions using numerical word-level representations (i.e., *word vectors*). These function as continuous *measures* of the information encoded in text, behavior, and brain data that allow for quantitative comparisons across these often incommensurate data types. Furthermore, because the representations are at the individual word level, they can be directly probed using widely available word ratings (norms) such as those we collate in *psychNorms*.

Our analyses rely on 10 text, 10 behavior, and 6 brain representations, and 292 word norms grouped into 27 norm categories (see Tables 1 and 2 for details). For our purposes, we subset each representation to a specific vocabulary. Specifically, for a given representation $i$, we take the intersection of its original vocabulary $V_i$ with the union of: (a) all the norm vocabularies $V_{\text{norm},n}$, (b) behavior embedding vocabularies $V_{\text{behavior},h}$, and (c) brain embedding vocabularies $V_{\text{brain},j}$. The resulting vocabulary $V_i'$ is defined as:

Table 1: Text, behavior, and brain representations (*trained as part of this research).

| REPRESENTATION | Description |
|---|---|
| fastText CommonCrawl | fastText architecture Mikolov et al. (2018), trained on CommonCrawl. |
| GloVe CommonCrawl | GloVe architecture Pennington et al. (2014), trained on CommonCrawl. |
| LexVec CommonCrawl | LexVec architecture Salle et al. (2016), trained on CommonCrawl. |
| fastText Wiki News | fastText architecture Mikolov et al. (2018), trained on Wikipedia 2017, UMBC webbase corpus and statmt.org news. |
| CBOW GoogleNews | CBOW architecture Mikolov et al. (2013) trained on the Google News. |
| fastTextSub OpenSub | fastText subword architecture Mikolov et al. (2018) trained on the Open-Subtitles corpus Van Paridon & Thompson (2021). |
| GloVe Wikipedia | GloVe architecture Pennington et al. (2014) trained on Wikipedia 2014. |
| spherical text Wikipedia | Spherical text architecture Meng et al. (2019) trained on Wikipedia 2019. |
| GloVe Twitter | GloVe architecture Pennington et al. (2014) trained on Twitter. |
| morphoNLM | Recurrent neural network architecture fine-tuned on morphological informative examples Luong et al. (2013). |
| norms sensorimotor | Ratings of 6 perceptual modalities and 5 action effectors Lynott et al. (2020) |
| SGSoftMax[In/Out]put SWOW* | [Cue/Response] vectors from Skip-gram softmax architecture (as in, e.g., Goldberg & Levy, 2014) trained on SWOW (De Deyne et al., 2019). |
| PPMI SVD SWOW* | Positive pointwise mutual information (PPMI) followed by singular value decomposition (SVD) of the SWOW cue-response frequency matrix (following, e.g., Richie & Bhatia, 2021; **?**). |
| PPMI SVD EAT* | PPMI followed by SVD of the Edinburgh Associative Thesaurus (EAT, Kiss et al., 1973). |
| SVD similarity relatedness* | SVD of a similarity matrix of aggregated and normalized similarity and relatedness judgment datasets [1] (and in the supplementary materials). |
| feature overlap | Cosine similarity matrix of overlapping feature frequency percentages between cue pairs in a feature listing task Buchanan et al. (2019) |
| THINGS | Neural network with softmax output trained to predict odd-one-out judgments of image triplets (Hebart et al., 2020). |
| experiential attributes | Human ratings on 65 attributes comprising sensory, motor, spatial, temporal, affective, social, and cognitive experiences (Binder et al., 2016) |
| eye tracking | Gaze patterns while reading for 7 datasets Hollenstein et al. (2019). |
| EEG text | Electrode measures while reading sentences (Hollenstein et al., 2018). |
| EEG speech | Electrode measures while listening to sentences (Broderick et al., 2018). |
| fMRI text hyper align | fMRI recordings while reading sentences (Wehbe et al., 2014), preprocessed by (Hollenstein et al., 2019) and hyper-aligned* across individuals. |
| microarray | Neuron-level recordings while listening to sentences |
| fMRI speech hyper align | fMRI recordings while listening to natural sentences (Brennan et al., 2016), preprocessed by (Hollenstein et al., 2019) and hyper-aligned* across individuals. |

$$V_i' = V_i \cap \left( \bigcup_n V_{\text{norm},n} \cup \bigcup_h V_{\text{behavior},h} \cup \bigcup_j V_{\text{brain},j} \right)$$

We do this for three reasons. First, it reduces the most numerous (text) vocabularies to a computationally feasible subset for representational similarity analysis (RSA, Section 3.2). Second, it focuses the analyses on a more psychologically relevant set of words—relevant in the sense that they are words that psychologists and neuroscientists have deemed suitable enough for inclusion in their data collection efforts. Finally, it ensures a more controlled comparison between representations by constraining their vocabularies to a more common subset.

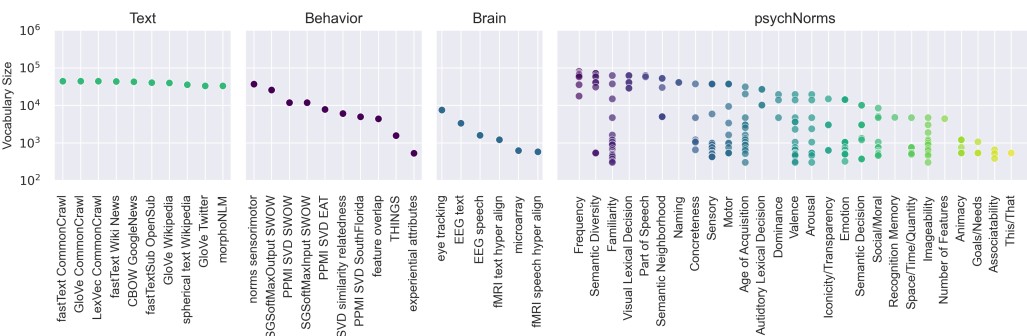

Figure 1: An illustration of the size of the vocabularies (y-axis, log-scaled) for each representation and norm (x-axis, grouped into higher-level categories) used in our analyses. The representations have been grouped into each data type (text, behavior, and brain).

Figure 1 illustrates the vocabulary sizes in log space. Starting from the left, the text representations reflect the largest vocabularies, with between $10^4 - 10^5$ words (following subsetting). Given text's dominance as a data source for training word representations, we were able to obtain a diverse set of high-quality *pretrained* representations from publicly available sources (see Table 1).

The behavior representations vary considerably in their vocabulary sizes, with the smallest (*experiential attributes*) on par with the smallest brain representations and the largest (*norms sensorimotor*) approaching that of text. We use a mixture of out-of-the-box behavior representations and those we train ourselves. For the latter, we rely heavily on the *Small World of Words* (SWOW) dataset (De Deyne et al., 2019), which is the largest dataset of free associations available. It contains roughly 3.6 million associates to over 12,000 cues, and has been found to be an effective way to uncover semantic representations in humans (Aeschbach et al., 2024).

Turning to the brain representations, vocabularies tend to be one or two orders of magnitudes smaller. We draw on preexisting fMRI and EEG data from reading ([fMRI/EEG] *text*) and listening ([fMRI/EEG] *speech*) tasks, eye-tracking data from a reading task (*eye tracking*, Hollenstein et al., 2018)[2], and a promising novel dataset of neuron-level recordings obtained from tungsten micro-electrode arrays (*microarray*) during listening tasks (Jamali et al., 2024). Aside from standard preprocessing steps and (hyper-)alignment of individual-level fMRI data (using the *HyperTools* Python package, Heusser et al., 2017), the brain data does not receive any further processing.

Finally, in order to measure the psychological content of the representations (via RCA), we needed a vast dataset of existing norms. Although norm (meta-)databases exist (e.g., Gao et al., 2023), there are (to our knowledge) no systematic literature searches for human-rated word properties. We thus screened 3,056 articles containing norm-relevant keywords (returning 181 norms) and combined the results with the largest preexisting norm metabase (SCOPE, 97 norms selected Gao et al., 2023) and a dataset of 65 human-rated experiential attributes (Binder et al., 2016). This resulted in a metabase of 292 unique norms, which we make available at github.com/[ANONYM]/psychNorms (and in the supplementary materials).

As illustrated on the right-hand side of Figure 1, these norms differ considerably both in the size of their vocabularies and the kinds of properties they seek to measure. To aid in interpretation of this diversity, we have manually grouped the norms (points) into higher-level categories (x-axis) (see Table 2). These categories include those that are popular in natural language processing settings (e.g., Frequency, Part of Speech, and Valence) as well as categories that have hitherto been relatively constrained to psycholinguistics (e.g., Space/Time/Quantity, Animacy, Goals/Needs).

---

[2]Although eye-tracking data is not typically considered brain data, we anticipated that the specific eye-tracking data used in this study, which was obtained from *reading tasks*, would be more closely linked to visual attention than, for instance, semantic relatedness judgments, which we view as more brain-like.

Table 2: Norm categories (*human-rated/behavioral norms).

| Category | Description |
|---|---|
| Frequency | (Log) frequency of word's occurrence in various text corpora. |
| Semantic Diversity | Measures word's polysemy or contextual diversity. |
| Familiarity* | Measures how well-known or familiar the word is. |
| Visual Lexical Decision* | Measures accuracy or response time during visual decision task with the word. |
| Part of Speech | The word's dominant grammatical category. |
| Semantic Neighborhood* | Network-style measures of the number and strength of the word's relationships with its neighbors. |
| Naming* | Measures accuracy or response time for word naming. |
| Concreteness* | Ratings of how concrete or abstract a word is. |
| Sensory* | Ratings of how strongly or easily the word is experienced through particular senses. |
| Motor* | Ratings of how much a word concerns body action or interaction. |
| Age of Acquisition* | Estimates of the age at which a word is learned. |
| Auditory Lexical Decision* | Measures accuracy or response time during auditory decision task with the word. |
| Dominance* | Ratings of the degree to which the word can be controlled. |
| Valence* | Ratings of how positive or negative a word is. |
| Arousal* | Ratings of the intensity of emotion or excitation evoked by a word. |
| Iconicity/Transparency* | Ratings of how much a word looks or sounds like what it means. |
| Emotion* | Ratings of how much a word reflect or elicits certain emotions. |
| Semantic Decision* | Accuracy or response time during semantic rating tasks. |
| Social/Moral* | Ratings of a word's relevance to social and moral dimensions. |
| Recognition Memory* | Recognition memory performance (hits minus false alarms). |
| Space/Time/Quantity* | Ratings of a word on spatial, temporal, and other quantitative dimensions. |
| Imageability* | Ratings of the ease with which a word can be imagined. |
| Number of Features* | Number of features listed for a word. |
| Animacy* | Ratings of how much a word is thinking, living, or human-like. |
| Goals/Needs* | Ratings of how much a word represents goals, needs, or drives. |
| Associatability* | Ratings of how quick and easy it is to thing of associations to a word. |
| This/That* | Proportion of times participants associated words with *this* versus *that*. |

## 3.2 REPRESENTATIONAL SIMILARITY ANALYSIS

We use representational similarity analysis (RSA) to compare the information encoded in the above representations. Developed within neuroscience (Kriegeskorte et al., 2008), RSA enables comparisons of representations from otherwise-disparate modalities (e.g., fMRI, EEG, similarity ratings) by leveraging the fact that the different dimensions may nevertheless contain information that seeks to distinguish a comparable set of mental states, stimuli, or other kinds of entities.

In our case, the entities being distinguished are words. Consequently, RSA measures the similarity between two matrices, $M_1$ and $M_2$, where each row $i$ represents a word, and each column $j$ reflects a measurement unit (dimensions). For the brain representations, these units may be voxels (fMRI) or electrode readings (EEG), whereas for text and behavior models, the units are often latent dimensions. RSA addresses the challenge of correlating these different units by transforming $M_1$ and $M_2$ into a common space. This transformation is achieved by calculating the (dis)similarities between the rows of $M_1$ and $M_2$, forming what is known as a *representational similarity matrix*, $S$. Following Lenci et al. (e.g., 2022)), we compute the *cosine* similarity matrices $S_1$ and $S_2$, as:

$$S_1 = \hat{M}_1 \cdot \hat{M}_1^\top \quad \text{and} \quad S_2 = \hat{M}_2 \cdot \hat{M}_2^\top,$$

where the hat notation $\hat{M}$ indicates that the rows of the matrices have been $L_2$ normalized. We then compute the similarity between the two representations by taking the Spearman correlation between the flattened upper triangles (excluding the diagonal) of $S_1$ and $S_2$.

### 3.3 Representational content analysis

Representational content analysis (RCA) is an approach to interpretable informational content *profiles* for abstract numerical representations. Although it leverages the well-established technique of probing from deep learning interpretability (see e.g., Belinkov, 2022), it differs from traditional probing applications in its scope, employing tens or even hundreds (as in our case) of targets to more holistically interpret the information encoded.

Our RCA implementation uses L2-regularized linear probing classifiers and regressors. We employ L2-regularization to mitigate issues such as multicolinearity, underdetermination, and over-fitting in high-dimensional settings. Following Hupkes et al. (2018), we use *linear* probes to avoid the risk of more flexible estimators learning features that do not reflect what is present in the original representations.

For numerical norms, we use the Scikit-Learn API's `RidgeCV` (Pedregosa et al., 2011). For binary and multi-class norms, we use the API's `LogisticRegressionCV`. Both estimators perform automatic (hyperparameter) tuning of the L2 penalty. This parameter—`alpha` in the case of `RidgeCV`, or `C` in the case of `LogisticRegressionCV` (equivalent to `1/apha`)—is selected from a grid of values ranging from $10^{-5}$ to $10^5$ (in `alpha` terms) with even spacing in log (base-10) space.

Generalization performance is measured via 5-fold nested cross-validation (Pedregosa et al., 2011), where the regression coefficients and L2 penalty parameter are fitted in an inner loop, and evaluated on separate test sets in the outer loop (following, e.g., Varma & Simon, 2006).

Finally, to ensure some minimum reliability for performance estimates, we do not probe in cases where the intersection of the representation and norm vocabularies results in a test set with fewer than 20 samples. This is important to keep in mind for Section 4.2, where, in a minority of cases, average performances are estimated from a reduced set of norms.

## 4 Experiments

### 4.1 Representations from text, behavior, and brain differ systematically, irrespective of learning algorithm

We begin by asking to what extent text, behavior, and brain data encode distinct information (research question (a)). Using representational similarity analysis (RSA), we compare the representations obtained from each data type (see Section 3.3 for details).

Figure 2 illustrates the results. Panel A presents a multidimensional scaling of the representational similarity space, and Panel B the pairwise similarity matrix. It is important to emphasize that each data type encompasses a diverse set of representations derived from different learning algorithms and sub-data-types (or sub-datasets) (see 3.1 for details). For instance, the text and behavior representations result from algorithms both from the *global matrix factorization* family (e.g., *PPMI SVD* SWOW, *SVD* Similarity Relatedness), *local context window* family (e.g., *fastText* CommonCrawl, *SGSoftMax Input* SWOW), and hybrids of both families (e.g., *GloVe* CommonCrawl).

Despite the diversity within data type, and some algorithmic commonalities between types (e.g., *fastText CommonCrawl*, *SGSoftMax Input SWOW*), we observe relatively clear clustering by data type (Figure 2), suggesting that the type of data has a more significant effect on representational structure than the choice of learning algorithm. Although some clustering based on the representation learning algorithm can be observed, the clustering by data is more pronounced.

To answer our research question, we find considerable differences between brain and behavior when compared to text (text-brain $\bar{\rho} = .09$, text-behavior $\bar{\rho} = .20$, where $\bar{\rho}$ denotes the mean Spearman correlation), with the similarities between the data types displaying lower values than those within (brain-brain $\bar{\rho} = .12$, behavior-behavior $\bar{\rho} = .22$, text-text $\bar{\rho} = .41$). Interestingly, the similarity

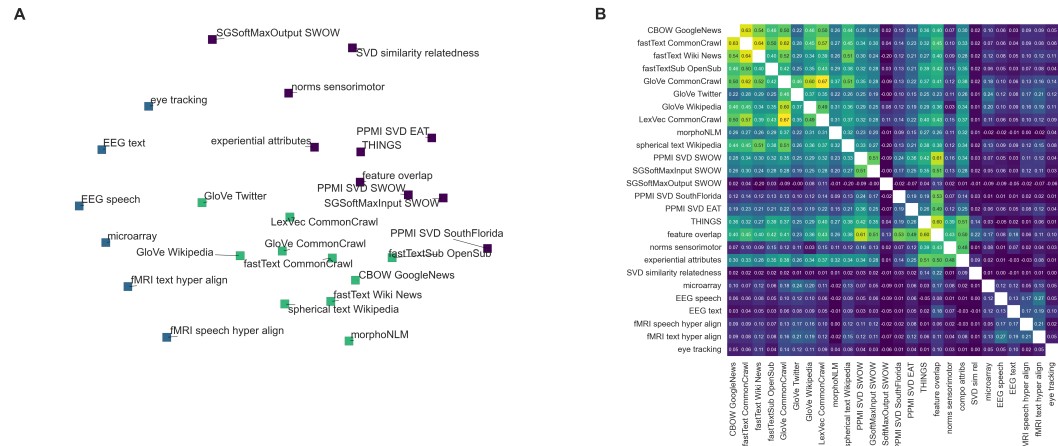

Figure 2: **A**: A 2-dimensional projection of the representational similarity space. The space was obtained by multidimensional scaling of the pairwise Spearman dissimilarity matrix between embeddings. Text = green, behavior = purple, brain = blue. **B**: A heatmap visualization of the pairwise Spearman similarity matrix.

between text and brain turns out to be .06 points higher than that between brain and behavior (brain-behavior $\bar{\rho} = .03$).

Ultimately, our analyses demonstrate the importance of data type in shaping representational similarity, with noticeable informational differences between text, behavior, and brain. We now move to characterizing these differences.

## 4.2 BEHAVIOR DATA CAN RIVAL TEXT IN PSYCHOLOGICAL BREADTH AND DEPTH

The last section revealed differences in the information encoded in text, behavior, and brain data. This raises the question: What is the *content* of these differences? This is important from the perspective of human-model alignment, where alignment on different dimensions will have varying implications for, for instance, a model's helpfulness, accuracy, or psychological plausibility. To address this question, we leverage our *psychNorms* metabase (Section 3.1) as targets in a representational content analysis (RCA, Section 3.3).

Figure 3 illustrates the average test performances of each representation[3] (rows) on each norm category (columns). Performance is measured via the coefficient of determination ($R^2$) for numerical norms, and McFadden's pseudo-$R^2$ for categorical norms (e.g. *This/That*, *Part of Speech* norms). We henceforth denote both measures with $R^2$.

Some interesting patterns can be observed. First, text and behavior appear to encode a broad range of psychological information. This is unsurprising in the case of text, which has been the dominant source for pretraining today's unprecedentedly human-like language models. Behavior, on the other hand, has garnered comparatively little attention in this regard. The representations are also derived from orders of magnitudes smaller training sets and possess more modest vocabularies (hence, smaller probe-training sets). Behavior's competitiveness with text is thus quite impressive.

Second, we detect scarce psychological information in brain. However, it is important to reiterate brain's limited vocabularies here. Furthermore, in many cases, the number of features (e.g., voxels, electrode readings) approaches the number of norm-labeled words (samples), making it all-the-more difficult to detect norm-signal in the brain data (i.e., even in cases where norm information is encoded). Nevertheless, in its present form, brain does not present a promising resource for human-model alignment.

---

[3]*feature overlap* and *experiential attributes* are dropped from remaining analyses due to, respectively, a vast number of missing values (words with no overlapping features were set to NaN), and an insufficiently large vocabulary.

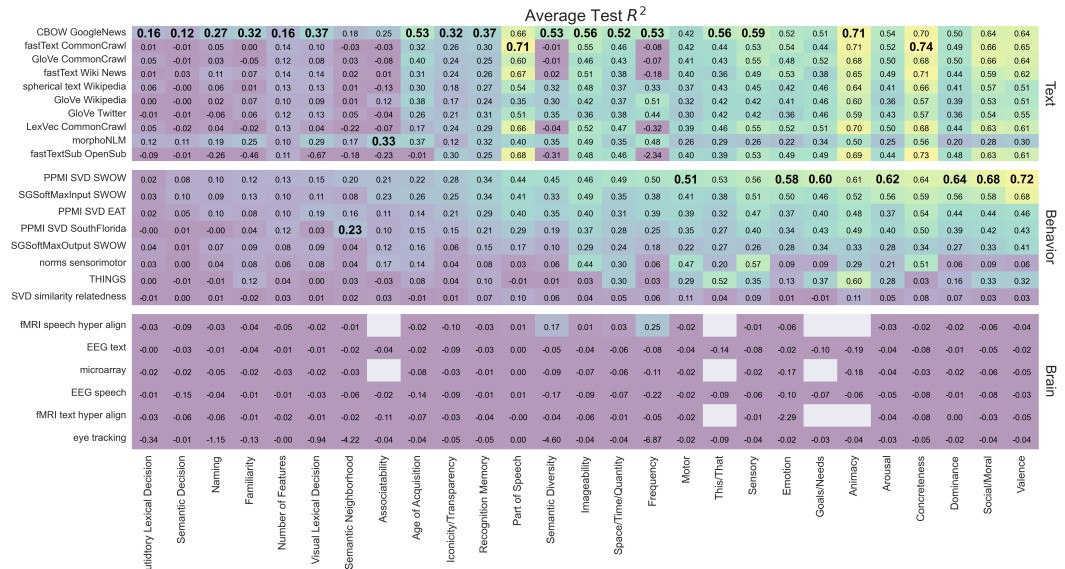

Figure 3: Average 5-fold cross-validation (pseudo-)$R^2$ test performance for text, behavior, and brain representations (rows, grouped) on 292 norms grouped into 27 norm categories (columns). Performances are aggregated by first taking the mean $R^2$ on each norm and then the median of the norm-wise (mean) $R^2$ for each norm category. Representations are ordered within each data type in terms of overall performance. Norms categories are ordered in terms of the performance of the top-performing behavior representation (*PPMI SVD SWOW*). Missing values are the result of an insufficient number of test samples.

Third, it appears that some norms are in general better-encoded than others across representations: namely, those on the right-hand side of Figure 3 versus those on the left. Although this may be explained in part by differences in norm reliability, it is also possible that certain norm-relevant information is especially hard to capture irrespective of data type. This latter explanation could indicate an avenue for future research seeking to capture remaining psychological information.

Fourth and finally, important differences can be observed between the best-performing representations from each type on certain norms. For instance, the best-performing text representations tend to outperform those of behavior by a considerable margin on *Part of Speech* (absolute difference in 90th percentile $R^2$, $|\Delta R^2_{90\text{th}}| = .26$), *Age of Acquisition* ($|\Delta R^2_{90\text{th}}| = .19$), *Visual Lexical Decision* ($|\Delta R^2_{90\text{th}}| = .14$), *Familiarity* ($|\Delta R^2_{90\text{th}}| = .13$, and *Concreteness* ($|\Delta R^2_{90\text{th}}| = .12$) norms. Of course, these superior performances may be (partially) attributable to the text representations' larger vocabularies (we control for probe-training set size and constitution in the next section, 4.3). The differences are nevertheless notable.

Conversely, the best-performing behavior representations perform comparatively strongly on *Dominance* ($|\Delta R^2_{90\text{th}}| = .09$), *Arousal* ($|\Delta R^2_{90\text{th}}| = .06$), *Motor* ($|\Delta R^2_{90\text{th}}| = .06$), *This/That* ($|\Delta R^2_{90\text{th}}| = .05$), and *Valence* ($|\Delta R^2_{90\text{th}}| = .05$) norms, relative to text. Given the behavior representations' smaller vocabularies, these higher performances can be seen as conservative estimates of what behavior may be able to contribute beyond text to human-LLM alignment.

All-in-all, our RCA provides a preliminary insight into the content of the differences between text, behavior, and brain. Having identified a surprisingly rich reservoir of psychological information in behavior, we now move onto the question of the extent to which behavior could complement text when it comes to human-model alignment.

### 4.3 Behavior captures unique psychological variance

The last section hinted that behavior may contain psychological information that text fails to capture. We now turn to the question of the *unique* (marginal) contribution of behavior on top of text. To

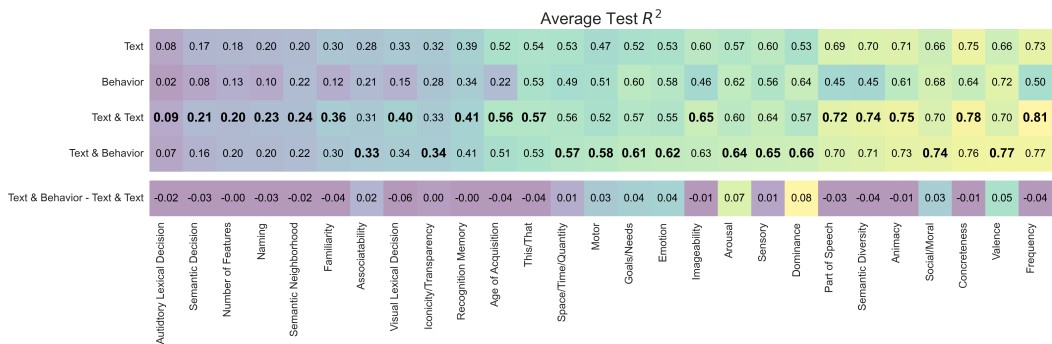

Figure 4: 5-fold cross-validation (pseudo-)$R^2$ performance for several text and behavior solo and ensemble representations (rows) on 292 norms grouped into 27 norm categories (columns). Performances are aggregated by first taking the mean (difference in) $R^2$ on each norm and then the median of the norm-wise (mean) $R^2$ for each norm category. Norms are ordered in terms of the performance of *Text & Behavior*.

investigate this, we perform an ensemble RCA, whereby we concatenate the top-performing text and behavior representations and measure the marginal increase in norm variance explained. We also subset all representation vocabularies to their collective intersection, meaning that the size and content of the probe's training set on any given norm is identical across representations.

Figure 4 illustrates the results. Specifically, we take the top-2 text representations from the previous section (*CBOW GoogleNews* and *fastText CommonCrawl*) and the top behavior representation (*PPMI SVD SWOW*). We then compare two main groups: *Text & Text*—in which we concatenate *CBOW GoogleNews* and *fastText CommonCrawl*—and *Text & Behavior*—in which we concatenate *PPMI SVD SWOW* with both *CBOW GoogleNews* and *fastText CommonCrawl*. We provide solo *Text* and *Behavior* baselines for reference.

The first thing to note is that ensembling tends to improve performance: on any given norm, it is either *Text & Text* or *Text & Behavior* in first place. However, neither *Text & Text* nor *Text & Behavior* is the unanimous winner. For instance, and as already hinted at in Section 4.2, *Text & Text* tends to outperform *Text & Behavior* on *Visual Lexical Decision* (absolute median difference, $|\tilde{d}| = .06$[4], Wilcoxon signed-rank $p < .001$), frequency-related norms (*Age of Acquisition*: $|\tilde{d}| = .03$, $p < .001$, *Familiarity*: $|\tilde{d}| = .03$, $p < .001$, *Frequency*: $|\tilde{d}| = .03$, $p < .001$), and *Semantic Diversity* ($|\tilde{d}| = .04$, $p < .001$) .

*Text & Behavior*, on the other hand, tends to perform better on affect-related norms (*Dominance*: $|\tilde{d}| = .08$, $p < .001$, *Arousal*: $|\tilde{d}| = .07$, $p < .001$, *Valence* $|\tilde{d}| = .06$, $p < .001$, *Emotion*: $|\tilde{d}| = .04$, $p < .001$), agency-related norms (*Goals/Needs*: $|\tilde{d}| = .03$, $p = .01$, *Motor*: $|\tilde{d}| = .04$, $p < .001$), and *Social/Moral* ($|\tilde{d}| = .03$, $p < .001$) norms.

Ultimately *Text & Behavior* (descriptively) outperforms *Text & Text* on 11 out of the 27 norm categories. Some of these categories (e.g., affective, agential, *Social/Moral* are likely crucial for human-LLM alignment, though their relevance will, of course, vary depending on one's ultimate alignment goals.

## 5 DISCUSSION

This article began by asking whether behavior and brain data could help in measuring and increasing human-LLM alignment (beyond text). We showed that behavior and brain representations encode information that differs from that of the text representations (Section 4.1). Drawing on our *psych-Norms* metabase and RCA, we probed these representations to reveal rich, interpretable psycholog-

---

[4]These numbers may differ slightly from those in Figure 4 due to differences in the level of mean aggregation at which the median was taken (fold-level means for Wilcoxon versus norm-level means for Figure 4).

ical profiles, with behavior outperforming text on several dimensions (e.g., *Dominance*, *Arousal*, Section 4.2). Motivated by evidence suggesting psychologically important differences between text and behavior, we carried out an ensemble RCA to reveal significant improvements from ensembling behavior with text on affective, agentic, and *Social/Moral* dimensions.

Our findings have important implications. The revealed differences in informational content can conceivably be exploited for human-LLM alignment. Consistent with the current practice of pre-training on text and fine-tuning on human behavior, our findings suggest that LLMs that are trained on multiple sources of data—specifically, text and behavior—are well-equipped to cover a larger number of dimensions relevant to human emotion, agency, and morality. Moreover, our RSA and RCA findings can be used to better understand the contents of behavioral datasets already used in the evaluation of language models—for instance, textual similarity judgements (e.g., Cer et al., 2017, dataset), sentiment judgments (e.g., Socher et al., 2013, dataset), and, more prospectively, free associations (Thawani et al., 2019; Abramski et al., 2024). Our analyses provide insight into both the content of these datasets, and how they *relate* to each other. We view this as crucial to improving our understanding of what is being evaluated or optimized for in such cases (Burden, 2024).

Our work has several limitations. First, it is foundational with respect to the entitled goal of *improving human-LLM alignment*: Although we demonstrate that behavior could *in principle* complement text in work seeking to measure or increase human-LLM alignment, we do not demonstrate this *in practice* (i.e., with the latest, SOTA LLMs). Nevertheless, the work provides hints at how this may be done—for instance, via RSA, RCA, or fine-tuning of the weights of SOTA models using behavior data (provided those weights are open, see Wulff et al., 2024)—and methods for comparing and aligning data from different modalities are not in short supply (see, e.g., Sucholutsky & Griffiths, 2023).

Second, our approach does not allow for perfectly controlled representational content comparisons. As mentioned in Section 4.2, although better probing results *may* signal the encoding of more norm-relevant information, they may also reflect larger probe-training set sizes. These issues can be alleviated by subsetting to the same vocabulary across comparison conditions (as we do in Section 4.3). However, this will naturally reduce the probe's sensitivity to norm-relevant signal due to the decrease in the training set size from subsetting.

One final limitation concerns our brain data, in which we detect scant evidence of psychological information. Although this may simply be due to the brain representations' small vocabularies, it could also be that brain is poorly suited to word-level analyses such as ours. After all, the brain data was collected during sentence-level tasks, meaning that word-level representations had to be extracted via relatively crude heuristics (e.g., a four-second hemodynamic delay offset) and averaging across contexts (Hollenstein et al., 2019). We would thus caution against drawing strong conclusions against other brain data formats (e.g., github.com/brain-score/language) on these bases.

## 6 CONCLUSION

In this work, we investigated behavior and brain data as prospective complements to text for measuring and improving human-LLM alignment. We found that behavior, in particular, captures psychological information to a breadth and depth rivalling that of text, and also captures unique psychological variance on certain dimensions. Our work thus contributes to a growing body of research (e.g., Bai et al., 2022; Kennington, 2021; Abramski et al., 2024) suggesting behavior as an important complement to text in LLM-training and evaluation pipelines, with the potential to improve LLM helpfulness, accuracy, and psychological plausibility.

## 7 REPRODUCIBILITY STATEMENT

Code and data for reproducing the analyses in this paper can be found in the supplementary materials, and will be made publicly available on GitHub upon publication. Anonymized GitHub links present in the paper will be de-anonymized for the camera-ready version.

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
