# OpenReview forum: "Probing the contents of text, behavior, and brain data toward improving human-LLM alignment"
_ICLR.cc/2025/Conference — ICLR 2025 Conference Withdrawn Submission_

### Official Review · Reviewer_7ZXT · 2024-11-04

**Soundness:** 2
**Presentation:** 2
**Contribution:** 3
**Rating:** 5
**Confidence:** 4

**Summary:**

The study conducts a similarity comparison of 10 text, 10 behavior, and 6 brain representations, presenting significant and consistent differences among these data types. The findings suggest that behavior data could be a valuable addition to text in LLM training, with the potential to improve LLM performance and human alignment.

**Strengths:**

I find the idea of the paper very interesting! I always wanted to explore word representations from different data sources, like behavior and brain. While the approach here is largely theoretical, it lays a foundation for future studies to explore these ideas in practical applications with widely used LLMs. Moreover, the Psychnorms metabase (if presented and described comprehensively) is a valuable contribution since it collects diverse behavioural word properties.

**Weaknesses:**

Overall, the paper is engaging and easy to follow, but certain points lack comprehension and would benefit from additional clarifications. As someone familiar with many of the datasets used, I find the presentation of these resources somewhat incomplete, and I am not sure how reliable are the results/claims of this work. Providing further clarifications and adding an Appendix describing the content and preprocessing of the datasets, would be a good start to address some of the following concerns.

Starting from the data collection and preprocessing:

PsychNorms is the metabase of what you call “behavioral word properties”. I would suggest providing further details of the metabase statistics and what it contains, the total size of the vocabulary, and a brief discussion on the norms, their overlap, etc. Table 1 provides a high-level overview but it would enhance the paper's clarity to provide further details. For example, for the THINGS database, it would be helpful to clarify that it is a "neural network trained on similarity judgments for real-world images". I recommend creating an Appendix and adding there additional information per representation.

Regarding the “brain” representations, it seems you used the CogniVal vectors provided by Hollenstein, 2019.  Did you use the Harry Potter, Alice and Pereira words? Please provide further details in the Appendix. The CogniVal database provides 100, 500, and 1000 random voxels. How do you make sure they indeed encode semantic information and not random brain signals? I suggest that to obtain word-level representations from neuroimaging data, the authors consider applying a Gaussian distribution over the fMRI voxels at the specific timestamp corresponding to each word, as demonstrated in [this study by Li et al., 2023](https://openreview.net/pdf?id=ZobkKCTaiY). Alternatively, word-level fMRI datasets are available, such as [Mitchell et al., 2008](https://www.science.org/doi/10.1126/science.1152876) and [Anderson et al., 2013](https://direct.mit.edu/tacl/article/doi/10.1162/tacl_a_00043/43388/Visually-Grounded-and-Textual-Semantic-Models).

Additionally, the details of processing the representations remain unclear. Did you perform any additional training on the vectors? The asterisk in the Figure 1 caption suggests, "*trained as part of this research" but no further information is provided. Moreover, the supplementary materials include only the *psychNorms* dataset and some notebooks, but without output examples, making it difficult to understand your implementation. How did you address the dimensionality or feature differences across modalities? For instance, CogniVal provides different feature dimensions for each dataset within fMRI, EEG, and eye-tracking data (e.g., *ucl_scaled.txt* has 5 features, *gecoscaled* has 7, and *dundeescaled* has 14). Did you use PCA or any other technique for dimensionality reduction?

Finally, it would be helpful to indicate the final vocabulary size for each representation, considering it is restricted to the intersection with the combined norm, behaviour, and brain vocabularies.

Regarding the methodology/results:

- Lack of clarity: RCA is not fully comprehensive. I recommend the authors to clarify how they evaluate and make sure Representational Content Analysis works as it should. Is it dependent on dimensionality variations? It is also not clear how you identify and measure the psychological information in the representations. I suggest providing further details about your implementation in the Appendix.

- Overclaiming: You need to soften your claims in some parts of your work regarding how the brain does not present a promising resource for human-model alignment. This finding might be affected by the random voxel representations you used from the CogniVal dataset. There seems to be a data type bias, probably as a result of the preprocessing that the data creators used to create the representations, so this is why the type of data has a more significant effect on representational structure than the choice of the learning algorithm.
This is briefly mentioned in lines 318-319 but is not highlighted as much as it should be.

The titles(claims) of subsections 4.2, and 4.3 do not seem to fully represent the content of the sections (figures and analysis of results). For 4.2, since you use the psychNorms metabase as targets there might be a bias in text&behavior compared to text&text? Also, it seems to me that the title of this subsection is misleading: 4.2 BEHAVIOR DATA CAN RIVAL TEXT IN PSYCHOLOGICAL BREADTH AND DEPTH. Why breadth and depth? The difference is approximately 0.1 between the text and behaviour data and only on 7 out of the 27 norm categories. Regarding 4.3, the main finding is that “Text & Behavior” (descriptively) has higher similarity than “Text & Text”, but the title is “4.3 BEHAVIOR CAPTURES UNIQUE PSYCHOLOGICAL VARIANCE” (probably this refers to the "Text & Behavior, on the other hand, tends to perform better on affect-related norms"?).

Minor: Both subsections start with “The *last* section revealed/hinted that..”. It is confusing to the reader what you refer to as last. If you refer to the previous subsections, I would suggest replacing *last* with *previous*.

**Questions:**

1. Thanks for providing the supplementary material, but since the data folder is partially empty it is hard to reproduce your work. It would be more helpful to actually see the notebook outputs, for example, what is the len(name_combs)? It would help the reader to get an estimate of how many final representations you compare. Figure 1 is the original size of the vocabularies, but you only use the overlap to run the experiments and present your results. Could you please provide further dataset/representation details or motivation as to why you have decided not to do so?

2. For people not familiar with the psychological norm datasets used, there is confusion between the behavioural and psychological norms. Could you please explain how “the behavior representations encode psychological information of equivalent or even superior reach and quality in comparison to their text-based cousins”? How do you measure that representations encode a broad range of psychological information?

4. By reading your paper it is not clear what is included in what behavioral dataset. Could you explain what is this out-of-the-box behavior representations in lines 186-187: “We use a mixture of out-of-the-box behavior representations and those we train ourselves”.

5. There is also a need to further explain the footnotes 3 and 4. I suggest having further details of the data and your implementation in the Appendix, as this could help handle those obscurities.

6. Since you already use the THINGS dataset, was there a reason for not using the fMRI, EEG, or MEG representations from https://things-initiative.org/ ?

7. Did you try to visualize some examples from your results as a qualitative check? I would recommend visualizing concrete word examples in a t-sne plot so that the readers can actually get an illustration of the correlation between the representations. For example, the word “x” is present both in the word embeddings, the Psychnorms, and the brain representations. Visualizing some examples can also help in the approximation of the relation between the representations and their data types.

8. It seems to me that in some parts, there is no clear distinction between correlation and performance. For some claims you should consider replacing “outperforms” or “superior/higher performances” with “there is a higher correlation” or “higher similarities”.

8. Minor improvements for paper clarity:

- Please fix the references throughout the text to avoid having **?** instead of the corresponding reference or citation.
- Please rephrase or further clarify lines 101-102: “Furthermore, because the representations are at the individual word level, they can be directly probed using widely available word ratings (norms)”.

---

> ### Author Response · Authors · 2024-11-19
>
> Thank you for your extensive, constructive feedback.
>
> Data collection and processing:
>
> We will make sure to include more information about data collection and processing in the Appendix in future versions of the work. We used the Harry Potter and Alice words as processed by the authors of CogniVal. We will look into methods for improving this processing; thank you for the suggestion. Finally, the code used to train the word vectors can be found in the data_prep directory in the Supplementary Materials. We will include more details in future versions of the paper’s Appendix.
>
> Methodology/results:
>
> We will provide further information on RCA in the Appendix in future versions. Regarding dimensionality, we previously ran analyses ensuring that the general pattern of results is robust to dimensionality. We will make sure to include these in future versions of the Appendix.
>
> Thank you for pointing out the potential data type bias due to common processing by the CogNival authors. We will run further analyses looking into the impact of brain data processing and possible bias. We will also consider including more brain datasets from other sources, as you suggest (thanks for the source suggestions!).
>
> Regarding the potential bias from evaluating Text & Behavior versus Text & Text on psychNorms, we would argue that this is more of a feature than a bug. If behavior representations are better aligned with behavioral norms that currently are the best available measures of important psychological dimensions, this is a good thing from an alignment perspective. This would change, of course, if we had other ways to accurately measure these dimensions without using behavioral response formats. Concerning the section title, ‘4.2 BEHAVIOR DATA CAN RIVAL TEXT IN PSYCHOLOGICAL BREADTH AND DEPTH’, we did not mean to imply that behavior is superior to text across the board– a good synonym for ‘rival’ in this context would be ‘match’ or ‘achieve comparable performance.’ Regarding 4.3 BEHAVIOR CAPTURES UNIQUE PSYCHOLOGICAL VARIANCE, we feel that this is in line with our results since Text & Behviour does significantly outperform Text & Text on several dimensions (Dominance, Arousal, Valence, Emotion etc.).
>
> To answer your questions:
> 1. Missing data can be downloaded by running `psychProbing/code/download_data.py` (as described under ‘Environment Setup` in the README). We will make sure to include notebook outputs in future data sharing.
> 2. Thanks for pointing this out. The behavioral norms used in psychNorms are some of the best available measures of the psychological dimensions we are seeking to measure. We show that ther representations encode a broad range of psychological information by demonstrating that these dimensions can be predicted by the probe in the RCA. We will be sure to make this point clearer to readers that are not familiar with psychological norms in future work.
> 3. By ‘out-of-the box’, we mean that we did not have to train the representations ourselves. This was, for instance, the case for THINGS, sensorimotor norms, and experiential attributes. We will make this clearer  and provide examples.
> 4. We will move these to the Appendix, where we can provide more extensive explanations as you suggest.
> 5. We felt we already had a sufficiently large number of brain representations. However, given the potential bias you mentioned from using the same CogNival-processed representations, we will make sure to investigate these further (and other potential data sources).
> 6. This is a really nice idea, thanks!
>  7. We only use ‘correlation’ in the context of RSA (where we are indeed taking correlations), and ‘performance’ in the context of RCA (where we are measuring out-of-sample performance). We thus feel that we are using the terms appropriately in their respective contexts.
> 8. Thank you for the corrections/suggestions for clarification; we will fix these.

---

### Official Review · Reviewer_yW6o · 2024-11-04

**Soundness:** 1
**Presentation:** 2
**Contribution:** 2
**Rating:** 3
**Confidence:** 3

**Summary:**

This paper investigates differences in text, behavior, and brain vector representations. For the text representation, the authors use or train fastText, GloVe, and CBOW word embeddings. For the behavior and brain representations, the authors leverage various behavioral datasets and brain datasets, such as EEG data. The authors also construct human-rated word properties across 27 categories, collectively named psychNorms. Experimental results show that these representations tend to cluster based on representational structure rather than the learning algorithm. Using psychNorms, the authors further explore relationships between specific categories in psychNorms and each representation by measuring the R^2 score.

**Strengths:**

The main strength of this paper lies in its exploration of interesting findings across multiple types of representations. The concept is sound and straightforward, with potential applicability to other modalities.

**Weaknesses:**

One primary concern involves overclaiming, particularly regarding large language models (LLMs). The authors use fastText, GloVe, and CBOW word embeddings and refer to these as LLMs. However, while definitions of LLMs vary, these models are generally not considered large by 2024 standards. One paper, for instance, suggests that models like GPT-3, which can solve few-shot tasks through in-context learning, meet the LLM criteria, whereas smaller models like GPT-2 do not [1]. Additionally, in the second paragraph of the Introduction, the authors mention “optimizing for next-token prediction” as a training method of LLMs, which does not accurately describe fastText, GloVe, or CBOW. Thus, the results presented here may not reflect human-LLM alignment as suggested.

[1] Zhao, Wayne Xin, Kun Zhou, Junyi Li, Tianyi Tang, Xiaolei Wang, Yupeng Hou, Yingqian Min et al. "A survey of large language models." arXiv preprint arXiv:2303.18223 (2023).

**Questions:**

- Who annotates the psychNorms metabase? Are they psychology experts, and how are the categories defined?
- What is the significance of the finding that the similarity between text and brain representations is 0.06 points higher than that between brain and behavior? Why is this an interesting result?
- How would multi-modal models that jointly incorporate text, behavior, and brain representations affect alignment between modalities? Such models might make alignment easier to detect since they learn these relationships directly.
- It would be helpful to highlight the results more prominently in the figures. Due to the current size, they are difficult to interpret and understand the results.
- There are missing elements in Table 1, such as “?” symbols and Footnote 1.

---

> ### Author Response · Authors · 2024-11-19
>
> Thank you for your feedback. To clarify, we do not refer to embeddings such as fastText, GloVe, or CBOW as LLMs. To avoid this misinterpretation of our approach, we were careful to refer to the embeddings as ‘representations’ or ‘measures’ of each data type (text, behavior, and brain) throughout the paper to distinguish them from the models that we ultimately hope to improve the alignment of LLMs.
>
> To answer your questions:
> 1. The psychNorms metabase predominantly seeks to measure lay people’s representations of words on certain psychological dimensions, for which there is often no objective ground truth. It is thus annotated by lay people, not experts. To clarify further, we have not collected this data; it has been taken from other (primary) databases.
> 2. It is somewhat surprising since one might expect that behavior data and brain data are more direct measures of people’s semantic representations than text, which would imply that they should be more similar to each other than to text. This is, of course, not true, given that text and brain are .06 points more similar than brain and behavior.
> 3. If one were to jointly derive a single representation from text, behavior, and brain data, this would probably improve the psychological richness of the representation. However, it is unclear how one would test how it would ‘affect alignment between modalities’ or make it easier to detect since they would be all mixed into a single representation. We may not fully understand this question.
> 4. Thank you for pointing this out; we will try to improve this in future versions of the work.
> 5. Thanks for pointing this out.

---

### Official Review · Reviewer_XtLg · 2024-11-04

**Soundness:** 2
**Presentation:** 2
**Contribution:** 2
**Rating:** 3
**Confidence:** 3

**Summary:**

This paper compares word embeddings with behaviour-based and brain-based word norms and tries to characterise their differences using a framework called representational content analysis (RCA) introduced by the authors. The authors first collect a large set of word representations from previous work based on text, behaviour, and brain. The authors then use representational similarity analysis (RSA) to measure the similarity of these word representations and find considerable differences between them. Finally, the authors propose RCA as an approach to interpret the differences.

**Strengths:**

The paper studies an interesting problem of comparing and contrasting word representations derived from text, behaviour, and brain. There has not been much work along this direction, to the best of my knowledge. In this sense, the work may provide new insights to the research community.

**Weaknesses:**

I have two major concerns with the work. First, there is a gap between the initial motivation of the work, which is human-model alignment, and the approach taken, which affects the soundness of the work assuming that human-model alignment is the ultimate goal of this work. This is because the paper focuses on comparing word representations derived from text with those derived from behaviour or brain. However, the behaviours of LLMs are probably dependent much more on the parameters of the deep neural network architecture than static word embeddings. It is not clear to me how the findings from this paper that characterize the difference between word embeddings and behaviour-based or brain-based word representations can be used for human-LLM alignment. If the authors could elaborate on how research outputs from this work can be applied for human-LLM alignment, it would strengthen the motivation.

Second, the paper is not easy to follow, which makes it hard to judge its soundness, especially for the proposed RCA. It seems the authors expect the readers to be familiar with existing behaviour representations and brain representations such as what they represent and how they are derived. The authors also expect readers to be familiar with RSA. Also, for the newly proposed RCA, it seems there is not any formal definition of this method. The description in Section 3.3 focuses on implementation details, but what does RCA take in as its input and what does it produce as its output?

**Questions:**

- It is stated that prospective data for human-model alignment can be grouped into three types: text, behaviour, and brain. Is this a widely adopted understanding? Besides the cited paper Roads & Love 2023 (which, based on its title, does not seem to be related to human-model alignment), are there other references supporting this claim?
- In RSA, how is the transformation from M_1 to \hat{M}_1 learned or derived? Can you elaborate more on the intuition that it is based on the dissimilarities between the rows of M_1 and M_2?
-  What is the input to RCA? What is its output?

---

> ### Author Response · Authors · 2024-11-19
>
> Thank you for your feedback. As we acknowledge in the discussion, we agree that the gap you mention is a weakness of the work and that the main value comes in the form of novel data and comparison methods. We will make sure to emphasize this aspect in future versions of the work.
>
> To answer your questions:
> 1. The claim that the data can be grouped in such a way is, in our minds, a logical (not empirical) claim, which we did not feel warranted further justification. Whether it makes empirical sense to group the data in such a way (e.g., because the groups are sufficiently different from one another to justify the grouping) is a claim we test and find support for in our RSA.
> 2. As mentioned in the section ‘Representational Content Analysis,’ the hat notation indicates that the rows of M_1 have been L2 normalized (we mention this in the paper).
> 3. The input is the word vectors (features) and word norms (targets). The output is 5-fold cross-validation performances for each set of word vectors on each word norm.

---

### Official Review · Reviewer_cmJx · 2024-11-12

**Soundness:** 2
**Presentation:** 3
**Contribution:** 2
**Rating:** 3
**Confidence:** 3

**Summary:**

This paper is a novel effort at understanding how different data sources (particularly, brain data and behavioural data in addition to text data) can contribute towards human-LLM alignment. The paper also identifies norms for which behavioural data could be more appropriate than textual data.

**Strengths:**

- The paper identifies the need for non-text related data in brain-LLM alignment
- The paper unifies a number of different norm-categories, but also a large number of textual, behavioural, brain datasets for this purpose.
- The paper performs RSA and RCA to understand the utility of these categories, which are reasonable and appropriate metrics.

**Weaknesses:**

- My fundamental issue with the paper is the lack of training pipelines for evaluating these new datasets in a new RLHF or other model, which the authors also note. Unsure of the novelty here, beyond the data aggregation + comparison metrics. Should this paper be in the datasets and benchmarks track? For example, the authors ask 2 research questions, one of which is "b) are these differences useful from the perspective of human-model alignment" and I don't see this question answered in the paper, since no training pipelines were used to verify their claims about the actual utility of behavioural data. Even though they show that behavioural data can capture certain norms better than text data, it's unclear if this difference is significantly important for LLM-human alignment. Could the authors:
* Train an RLHF pipeline on the text+behavioural data and compare it to SOTA LLMs today. Start with a base LLM, train a reward model to rank responses based on the text + behavioural data, fine-tune the base model to produce responses that will be scored highly by the reward model using an RL algorithm such as Proximal Policy Optimization (PPO). Conduct quantitative and qualitative evals on the reward model as well as the final fine-tuned LLM . For example, quantitative evals could be to look at reward values in the RM and logit/surprisal/perplexity values in the policy-tuned LLM and compare them to the text-based RLHF baselines. Qualitative evals could look at reward values on N generations to the same input, as well as standard generations on the final LLM. The final policy-tuned LLM could also be evaluated on standard LLM benchmarking tasks to validate its performance based on, say, the affect-related norms/social norms. The SocKET Benchmark (https://arxiv.org/abs/2305.14938) could be appropriate here.

Minor:
 - This is unsurprising in the case of text, which has been the dominant source for pretraining today’s unprecedentedly human-like language models. (Please CITE)
- Some of these categories (e.g., affective, agential, Social/Moral are likely crucial for human-LLM alignment, though their relevance will, of course, vary depending on one’s ultimate alignment
goals. (Vague. Please CITE)

**Questions:**

N/A

---

> ### Author Response · Authors · 2024-11-19
>
> Thank you for your feedback. As we acknowledge in the discussion, we agree that the lack of training pipelines for evaluation in SOTA LLMs is a weakness and that the main value comes in the form of data aggregation and comparison. We will make sure to emphasize this aspect in future versions of the work.

---

### Note · Authors · 2024-11-19

**Comment:**

Dear reviewers, thank you for your feedback on our work. We agree with the general consensus that the main weakness of the work is that it fails to demonstrate (in practice) how the novel data could be used to improve SOTA LLMs. Given the overall response, we have thus decided to withdraw our submission. We nevertheless believe that foundational work such as ours seeking to understand the contents of prospective data types for training LLMs is important, even if it does not provide a direct path towards implementation. We will be sure to make use of your comments to improve future versions of the work. We have responded to your individual feedback below. Thank you!

**Withdrawal Confirmation:**

I have read and agree with the venue's withdrawal policy on behalf of myself and my co-authors.